# The Reversion of DNA Methylation at Coronary Heart Disease Risk Loci in Response to Prevention Therapy

Willem Philibert [1], Allan M. Andersen [1], Eric A. Hoffman [2,3], Robert Philibert [1,2,3] and Meeshanthini Dogan [1,3,4,*]

1 Department of Psychiatry, University of Iowa, Iowa City, IA 52242, USA; willem-philibert@uiowa.edu (W.P.); allan-andersen@uiowa.edu (A.M.A.); robert-philibert@uiowa.edu (R.P.)
2 Department of Radiology, University of Iowa, Iowa City, IA 52242, USA; eric-hoffman@uiowa.edu
3 Roy J. Carver Department of Biomedical Engineering, University of Iowa, Iowa City, IA 52242, USA
4 Cardio Diagnostics Inc., Coralville, IA 52241, USA
* Correspondence: mdogan@cardiodiagnosticsinc.com or meeshanthini-vijayendran@uiowa.edu

**Abstract:** Coronary heart disease (CHD) is preventable, but the methods for assessing risk and monitoring response rely on imprecise lipid-based assessments. Recently, we have shown that an integrated genetic–epigenetic test that includes three methylation-sensitive digital PCR assays predicts 3-year risk for incident CHD better than lipid-based methods. However, whether methylation sites change in response to therapies that alter CHD risk is not known. Therefore, we assessed methylation at these three incident CHD-related sites in DNA from 39 subjects before and after three months of biochemically verified smoking cessation, then analyzed the relationship between change in methylation at each of the sites to the change in smoking intensity as assessed by cg05575921 methylation. We found that, in those who quit smoking, methylation change at one CHD risk marker (cg00300879) was significantly associated with change in cg05575921 methylation ($p < 0.04$). We conclude that changes in incident CHD-related methylation occur within three months of cessation of smoking, a major risk factor for CHD. This suggests that the effectiveness of treatment of other CHD risk factors, such as high cholesterol, may be similarly quantifiable using epigenetic approaches. Further studies to determine the relationship of changes of methylation status in response to treatment of other CHD risk factors are indicated.

**Keywords:** coronary heart disease; smoking cessation; DNA methylation; cg05575921; epigenetics; treatment response; precision medicine



## 1. Introduction

Coronary heart disease (CHD) causes the deaths of eight million people each year [1]. CHD-related deaths are largely preventable, but effective prevention requires that risk for CHD be accurately determined. Current strategies for assessing risk for CHD begin with the determination of serum lipid levels, which are then incorporated into metrics such as the American Heart Association/American College of Cardiology Atherosclerotic Cardiovascular Disease risk calculator (ASCVD) or the European Systemic Coronary Risk Estimation (SCORE) assessment [2,3]. When indicated, the results of these metrics are used to guide response to therapy, such as the use of statins to reduce the patient's cholesterol level, or modifying lifestyle habits, such as smoking or poor diet, that are associated with increased risk for CHD [4]. Unfortunately, these approaches for quantifying CHD risk often fail, with up to 80% of all cardiovascular events that currently occur being otherwise preventable [5]. Although there are several reasons for these failures, two of the most commonly cited reasons are inadequate mechanisms for assessing risk and imprecise methods for monitoring the effectiveness of therapy in those who are deemed to be at risk.

In hopes of improving prevention and addressing these failures, we have recently introduced Epi + Gen CHD™, an integrated genetic–epigenetic test for assessing the 3-

year risk for incident CHD. As opposed to prior epigenetic methods that use arrays to measure methylation, Epi + Gen CHD™ uses three methylation-sensitive digital PCR (MSdPCR) assays to assess methylation status in key pathways. The resulting epigenetic information and the information from five genotype assays is then interpreted by an artificial intelligence (AI) algorithm status.

The adoption of MSdPCR for the methylation assessments is critical for the clinical translation of this AI-based approach. Array-based methods of assessing methylation are costly (>USD 200 each), need to be run in groups, are time-consuming (~1 week to complete testing), and, critically, are reference-dependent [6–8]. In contrast, MSdPCR methods are precise, relatively inexpensive to conduct, fast, and reference-independent [9]. As such, MSdPCR assays may be an ideal tool serving as the laboratory backbone of Precision Epigenetic Medicine approaches.

As implemented in the Epi + Gen CHD™ test, these assays appear to work well. In direct head-to-head comparisons, we have shown that the Epi + Gen CHD™ predicts CHD incidents within a 3-year window better than ASCVD on average, with >20% and around 10% greater sensitivity and specificity, respectively [10]. Still, like most NextGen prevention technologies, such as Cologuard™, the cost of this test is currently higher than that for existing lipid-based approaches. However, cost–utility analyses show that implementation of this method would not only save lives but decrease overall healthcare costs [11]. As such, this suggests that implementation of this or a similar integrated genetic–epigenetic technology could address the current shortcoming in quantifying risk for CHD.

However, the question remains whether this or a similar epigenetics-based approach could be used to monitor response to medical or behavioral therapy. Currently, the clinical response to elevated risk based on the ASCVD risk calculator or similar indices fits into one of two categories [3]. The first is the use of statins to reduce cholesterol levels. The second is lifestyle modifications that address contributing co-morbid medical conditions, such as high blood pressure, or lifestyle factors, such as poor diet and smoking, that are associated with increased risk.

Monitoring clinical response accurately is not easy. The main method through which the effectiveness of therapy is determined is by assessing serum lipid levels [3]. However, current clinical practice requires the patient to be fasting and implicitly assumes that the patients' dietary repertoire prior to sampling is reflective of their general dietary habits. Furthermore, although treatment with statins is associated with decreased cholesterol levels, the use of some, but not all, statins increases hemoglobin A1c (HbA1c) levels, the main biomarker for monitoring treatment of diabetes mellitus (DM) [12,13]. Since DM is also a risk factor for CHD, physicians must sometimes balance the effectiveness of cholesterol control with the control of glucose levels. Control of blood pressure and monitoring the effectiveness of lifestyle modifications is also essential. However, taking reliable blood pressures and assessing the frequency of undesirable social behaviors in a clinical setting can be a difficult, time-consuming proposition [14,15]. Taken together, these observations suggest the need for an easier to implement, more robust method for assessing progress in CHD prevention therapy.

Theoretically, if methylation at the three CpG loci interrogated by the three MSdPCR assays is both dynamic and predictive of risk, methylation at these loci should revert as a function of effective CHD prevention therapy. For example, with respect to statin therapy, as serum cholesterol decreases in response to statin therapy, one should expect the methylation at the CpG loci that map to cholesterol levels to change accordingly. Unfortunately, because most CHD prevention studies do not obtain DNA samples from both before and after treatment, we have not been able to determine whether statin therapy is associated with reversion of methylation at the three methylation sites. However, we recently completed a study of the effects of smoking cessation on pulmonary inflammation. In that study, subjects were offered USD 400 of financial incentive to quit smoking without the use of smoking cessation agents—in particular, nicotine replacement therapy [16]. In

total, 20 subjects completed 3 months of cotinine-verified smoking cessation and provided DNA samples before and after 3 months of biochemically verified smoking cessation.

Critically, smoking is one of the largest preventable causes of heart disease. Furthermore, unlike the cases for many other lifestyle or metabolic risk factors for CHD, we can reliably determine both the compliance with prevention (i.e., cessation) therapy by measuring serum cotinine and measure the change in risk factor (i.e., smoking) intensity by assessing DNA methylation at cg05575921 [17]. Therefore, as a first attempt to constrain the timing and effect size of possible epigenetic changes that might be expected in peripheral DNA methylation as a function of effective CHD prevention therapy, we examined DNA methylation at the three loci used in the Epi + Gen CHD$^{TM}$ before and after 3 months of cotinine-verified smoking cessation.

## 2. Materials and Methods

Study Approval: All subjects who participated in the study provided informed written consent. All procedures used in the study were approved by the University of Iowa Institutional Review Board (IRB201706713).

Study Participants: The 39 participants whose data are included in this study were part of a cohort of 67 subjects recruited in a series of advertisements seeking adult daily smokers, distributed to patients and staff at the University of Iowa Hospitals and Clinics [16]. Those subjects who were potentially interested in the study were invited to complete an online survey on their smoking habits. Those subjects who reported smoking more than 10 cigarettes a day and had at least 5 pack-years of lifetime consumption in the survey were then invited to participate in the smoking cessation protocol.

In brief, as part of the study to determine the effects of smoking cessation on pulmonary inflammation, subjects were offered USD 400 if they successfully quit smoking. Successful quitting was defined as a self-report of quitting smoking accompanied by serum cotinine values of less than 10 ng/mL at the first-, second-, and third-monthly clinical visit. Subjects were encouraged to stop "cold turkey" and to abstain from using standard smoking cessation treatments, particularly nicotine replacement therapy, to quit smoking. Subjects were offered a brief counseling session led by a research assistant at each study visit and a weekly phone call over the first month of the study. Subjects were considered treatment failures if they had serum cotinine values above 10 ng/mL at any time point or failed to attend any of the clinical visits [3]. Only 20 of the original 67 subjects completed all procedures, reported quitting smoking, and had serum cotinine values of <10 ng/mL at all three monthly clinic visits. Nineteen others who provided DNA for this study also completed all four visits but had serum cotinine values of >10 ng/mL at one or more visits.

Laboratory Measures: All subjects were phlebotomized at intake and during each monthly clinic visit to provide serum and DNA for the current study. Serum cotinine levels were determined by University of Iowa Diagnostic Laboratories under standard CLIA- compliant procedures. Relative change in DNA methylation at cg05575921, a well-established epigenetic indicator of smoking intensity, and three other sites used in the Epi + Gen CHDTM test were quantified by personnel blind to subject status [10,17]. Whole blood DNA from each subject at each time point (monthly meeting) was prepared as previously described [17].

DNA methylation at cg05575921 and the three methylation sites in the Epi + Gen CHD™ test (cg00300879, cg09552548, and cg14789911) was performed as previously described using proprietary methylation-sensitive, nested, digital primer probe sets from Behavioral Diagnostics and Cardio Diagnostics (Coralville, IA, USA) and droplet digital PCR reagents and machinery from Bio Rad (Carlsbad, CA, USA) [10,17]. In brief, 1 ug of DNA from each subject at study intake (baseline) and study exit (month 3) was bisulfite-converted using a Fast 96 Epitect Kit (Qiagen, Germany), with the resulting DNA being eluted using 70 μL of 10 mM Tris buffer (pH 8.0). A 3 μL aliquot of the resulting product was pre-amplified using the assay-specific pre-amplification mix, then diluted 1:1500 for the Epi + Gen CHDTM assay, or 1:3000 for the cg05575921 assay. After dilution, a 5 μL

aliquot containing approximately 10,000 amplicons—mixed with universal droplet digital PCR reagents and fluorescent primer probe sets specific to the cg00300879, cg09552548, cg14789911, and cg05575921 loci—was partitioned into droplets and then PCR amplified using a QX-200 Droplet Digital PCR system (Bio Rad) according to manufacturer's instructions. After amplification was complete, the number of droplets containing amplicons with at least one "C" allele, one "T" allele, or neither allele was then determined using a Bio-Rad QX-200 Droplet Reader, and the absolute ratio of methylated to total CpG methylation at each was determined by the proprietary Bio Rad Quantisoft™ software.

Statistical Analyses: All data were analyzed using the JMP Version 14 (SAS Institute, Cary, SC, USA) using standard general linear model equations [18]. Group comparisons of continuous variables were compared using T-Tests. Bivariate regression was used to analyze the relationship of changes of epigenetic-indicated smoking intensity (cg05575921) to change in cardiac methylation marker (cg00300879, cg09552548, and cg14789911) status.

## 3. Results

The clinical and demographic characteristics of the 39 subjects who completed all four clinic visits and whose data were used in this study are given in Table 1. In brief, they tended to be in the late 30s to early 40s in age, with a slight majority being male. All but two of the subjects were White.

**Table 1.** Demographic and clinical characteristics of the subjects.

| | **Quitters** | **Non-Quitters** |
|---|---|---|
| N | 20 | 19 |
| Age | $39.8 \pm 9.9$ | $45 \pm 10.2$ |
| Gender | | |
|     Male | 11 | 11 |
|     Female | 9 | 8 |
| Ethnicity | | |
|     White | 20 | 17 |
|     African American | - | 1 |
|     Other | - | 1 |
| Pack-Year Consumption | $22 \pm 9.6$ | $34 \pm 25$ |
| Cigarettes per day | $16 \pm 6$ | $19 \pm 13$ |
| Intake Cotinine (ng/mL) | $206 \pm 93$ | $278 \pm 135$ |
| Intake Methylation (%) | | |
|     cg00300879 | $52.9 \pm 9.1$ | $58.3 \pm 11.7$ |
|     cg09552548 | $30.0 \pm 11.7$ | $32.0 \pm 13.4$ |
|     cg14789911 ** | $89.7 \pm 9.3$ | $81.2 \pm 16.4$ |
|     cg05575921 | $57.1 \pm 22.1$ | $47.2 \pm 18.3$ |
| Δ Methylation (%) over 90 Days | | |
|     cg00300879 | $-0.7 \pm 1.6$ | $1.0 \pm 4.6$ |
|     cg09552548 | $0.1 \pm 0.8$ | $0.2 \pm 1.0$ |
|     cg14789911 | $-1.5 \pm 3.4$ | $-1.1 \pm 5.2$ |
|     cg05575921 | $-7.6 \pm 5.8$ | $-2.1\% \pm 5.5$ |

** nominally different at $p < 0.05$.

Only 20 of the subjects who completed all four clinic visits managed to quit smoking, as evidenced by negative cotinine values at all three monthly visits. There were no significant differences in cigarette consumption or serum cotinine values between those who quit and those who did not quit ($p > 0.05$ for both).

We determined methylation levels at each of the three CpG sites used in the Epi + Gen CHDTM test in each of the subjects at study entry and study exit ninety days later (see Table 1). Please note that because the set point of these methylation sites is genetically contextual and we did not determine genotype at the five sites used in this test, a direct comparison of the methylation values for those who quit versus those who did not quit is not possible. Nevertheless, in general, lower methylation values at cg00300879 and

cg09552548, but higher methylation levels at cg14789911, are associated with increased risk for incident CHD within three years [10].

Over the course of the study, methylation arithmetically increased at cg00300879 and cg09552548 and decreased at cg14789911 in those who quit smoking. However, using a categorical approach to classify smoking cessation, none of the changes at these three loci were significant. The subjects who were unsuccessful in quitting smoking manifested lesser degrees of change at the three loci over the 90-day period, all of which were also not significantly associated with categorical quitting status.

However, when considering these results, it is important to realize that, just as not all cases of CHD are equally severe, not all smokers consume the same number of cigarettes. Fortunately, the use of cg05575921 as a metric for change in smoking intensity permits the stable objective measurement of smoking intensity [17,19]. Recently, we have shown that changes in cg05575921 methylation in response to smoking cessation are also dose-dependent [16]. Therefore, in order to determine whether the changes in smoking intensity were related to the changes in methylation, we analyzed the relationship of the change of methylation at each of the cardiac markers to the change in smoking intensity as measured by change in cg05575921. As expected, even though some of the subjects showed objective evidence of decreasing the rate of smoking, there were no significant relationships between the change in smoking intensity and changes in methylation at any of the three cardiac-specific loci in the 19 subjects who did not completely quit smoking. However, in the group of 20 subjects who managed to quit smoking completely, after correction for multiple comparisons, there was a significant relationship between the smoking-cessation-induced reversion of methylation at cg05575921 to an increase in methylation at cg00300879 (Adj $R^2$ 0.26, $p < 0.04$, Figure 1C), with the changes in methylation at cg147989911 failing to achieve statistical significance (Adj $R^2$ 0.14, $p < 0.07$).

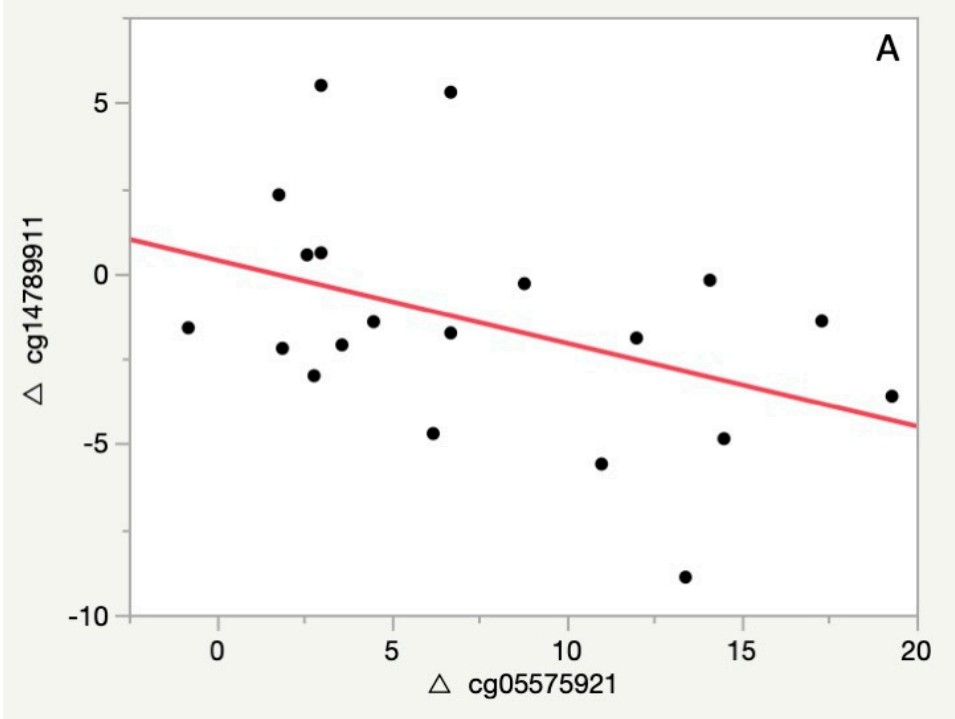

**Figure 1.** *Cont.*

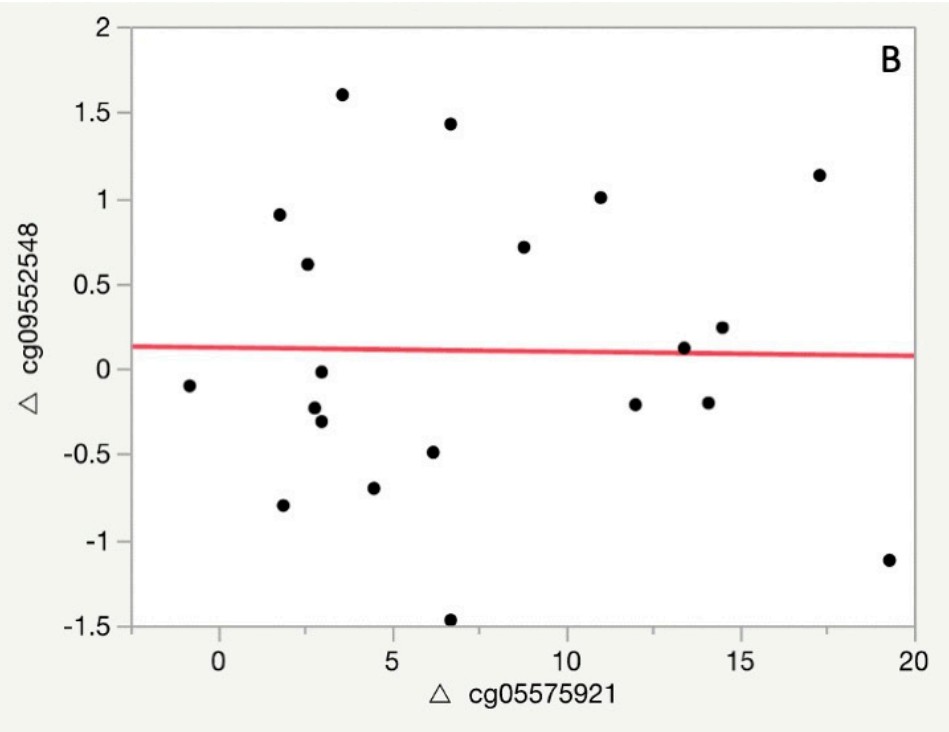

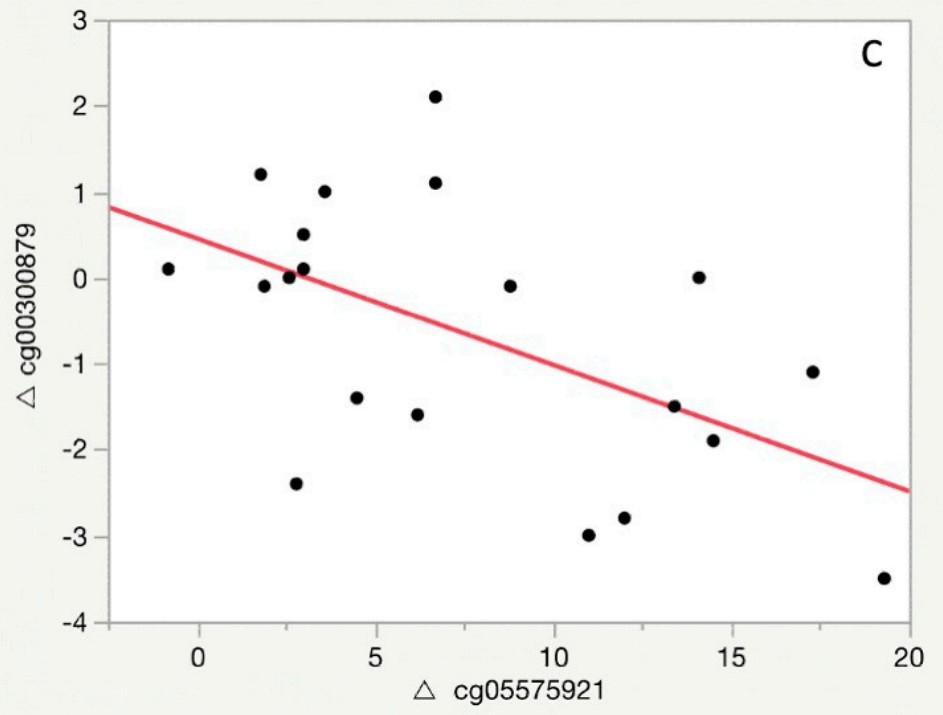

**Figure 1.** The relationship of change in increases in cg05575921 methylation seen in response to smoking cessation to changes in methylation at each of the three cardiac risk loci between study entry and study exit (3 months) in the 20 subjects who had biochemically verified smoking cessation. (**A**) is a plot of the change of cg14789911 with respect to the change of cg05575921. (**B**) is a plot cg09552548 with respect to the change of cg05575921 methylation. (**C**) is a plot of the change of cg00300879 with respect to the change cg05575921 methylation. The change illustrated in (**C**) is significant after Bonferroni correction (Adj $R^2$ 0.26, $p < 0.04$). A negative $\Delta$ indicates an increase in methylation at the cardiac risk marker site.

## 4. Discussion

In summary, the current results of this pilot study show changes at the CpG loci predictive of incident CHD in association with the biochemically verified treatment of a risk factor for CHD.

Although the sample size is small, we believe that the current study illustrates the potential for Precision Epigenetic Medicine—and, in particular, MSdPCR—for monitoring the effectiveness of prevention therapy for CHD. The current illustration of this potential future took advantage of several unique features of smoking. First, smoking is a clear, well-defined, large-effect risk factor (~1.5 fold) for CHD [20]. The effect size of other CHD risk factors, such as obesity, may be similar. However, many of these other risk factors are frequently co-morbid with others. As a consequence, testing the effect of treating these factors in isolation from other risk factors may be more difficult to perform. Second, there are clear biomarkers for smoking that are both categorical (e.g., cotinine) and quantitative (DNA methylation) in nature. This allows us to unambiguously determine whether a subject is quitting smoking and the degree of that change. Third, cigarette smoke is an exogenous risk factor that can be completely eliminated. In contrast, although it is possible to decrease serum cholesterol and glucose levels in those with hypercholesteremia and diabetes, respectively, completely eliminating cholesterol and glucose is neither desirable nor possible.

The finding of significant changes in cg00300879 makes intuitive scientific sense. The probe assesses methylation at a CpG site found in a candidate cis-regulatory element (cCRE) found in the promoter region of CNKSR1 (Connector Enhancer of Kinase Suppressor of Ras 1), a scaffold protein for receptor kinase signaling [21,22]. Changes in CNKSR1 gene expression are noted in numerous patent claims related to lung cancer, a disease which is highly associated with smoking [23,24]. CNKSR1 directly interacts with SASH1, a key protein in the interactome that mediates gene expression changes associated with atherosclerosis in smokers [25]. The changes observed in methylation (average 0.7%) at cg00300879 in this study are unlikely to be of immediate physiological consequence. However, if the reversion in the methylation continues to occur as a function of continued smoking cessation, they could increase to the point where they could conceivably be associated with changes in CNKSR1 gene and protein expression. Since the risk for CHD conferred by smoking cessation does not manifest for several years, and the full reversion of smoking-induced cg05575921 demethylation associated with smoking cessation takes several years (i.e., 5–10 years) in heavy smokers [17], it is likely that the reversion of methylation observed at cg00300879 will also continue for several years, albeit slowly, in those subjects who quit smoking. Demonstrating this definitively will require well-designed and powered longitudinal studies.

Conceivably, any change in risk for CHD could be useful in motivating smokers to quit smoking. Secondary to the social stigma associated with smoking, patients are reluctant to engage with their physicians about the hazards of smoking [26]. However, CHD prevention has no such social stigma, and recent work with biomarkers of hepatic injury has shown the benefit of physicians conveying similar information about the protective effects of alcohol cessation on liver function to patients with alcohol dependence [27]. Using potential effects of smoking cessation on CHD risk, alone or together with incentive-based programming methods for smoking cessation, could be a powerful motivator to persuade nicotine-dependent smokers to quit [16].

Unfortunately, we do not have information on the effect of smoking cessation on other CHD risk variables, such as serum cholesterol and hemoglobin A1c levels, in these subjects. This prevents us from understanding whether the changes in cg00300879 are best associated with changes in smoking or some other change in known cardiac risk factors. Smoking increases serum low-density lipoprotein (LDL) levels and is a risk factor for type 2 diabetes (T2D). Normally, serum LDL levels decrease as a function of smoking cessation and scaffold connectors for kinases such as CNKSR1 are known to preferentially interact with cholesterol-rich domains of the plasma membrane [28,29]. Therefore, it is

quite conceivable that the changes in serum cholesterol or some other risk factor could be better associated with the changes in CNKSR1 methylation. However, rigorously testing this hypothesis would be exceedingly difficult, as Verdugo and associates have shown that the interactome associated with smoking-induced atherosclerosis encompasses over 3000 genes [25].

The current findings have potential for improving CHD prevention in those with multiple risk factors. In particular, we believe that developing an epigenetic method of monitoring CHD risk may improve management of those with elevated cholesterol levels and subclinical or overt T2D. A conundrum for clinicians is the knowledge that statin-induced decreases in serum cholesterol levels are often associated with increases in HbA1c levels [12,13]. Overall, the risk/benefit ratio for the use of statins is favorable [12]. Whether this general benefit applies to all patients evenly is not known, because current algorithms cannot simultaneously consider changes in lipid and HbA1c levels. However, because each of the methylation markers maps differently to principle components of the methylation response associated with CHD, the change in overall risk as a consequence of changes in serum cholesterol and HbA1c levels can be assessed simultaneously by our integrated assessment tools [10]. For most patients, we believe that the added information by retesting methylation levels is unlikely to change risk management. However, for some—particularly those with genetic polymorphisms that alter HbA1c levels [30]—the added information could be valuable. We suggest that clinical studies of new cholesterol or glucose-lowering agents consider banking DNA from several time points in order to provide the biomaterial to test this hypothesis.

Finally, certain limitations of the study should be noted. First, the sample size of the study, particularly the group that quit smoking ($n = 20$), was rather small and consisted almost exclusively of those of European ancestry. Second, we only examined the loci used in the Epi + Gen CHD™ test. A suitably powered genome-wide approach could conceivably identify other markers even better suited to monitoring or quantifying the degree of change in risk for CHD conferred by the change in smoking behavior. Third, we only examined the subjects for three months. Ideally, it would have been desirable to assess the effects of smoking cessation on DNA methylation and cardiac health over a period of years.

In summary, we report that changes in smoking intensity are associated with significant reversion of CHD associated methylation. We suggest that further studies to better refine the time course of reversion, and, more generally, to determine whether Precision Epigenetic techniques can be used to guide this, and other forms of preventive cardiac care are indicated.

**Author Contributions:** W.P. conducted the cardiac methylation assessments, conducted the initial analyses, and wrote the first draft of the manuscript. E.A.H. and R.P. obtained the funding for the project. E.A.H., A.M.A., R.P., and M.D. contributed the writing of the final manuscript. All authors have read and agreed to the published version of the manuscript.

**Funding:** This work was supported by 5R01HL130883 (Eric Hoffman, PI) and R44CA213507 (Robert Philibert, PI).

**Institutional Review Board Statement:** All protocols and procedures used in this study were approved by the University of Iowa Institutional Review Board (IRB201706713).

**Informed Consent Statement:** Informed consent for molecular studies was obtained from all subjects in the study.

**Data Availability Statement:** The datasets used during the current study are available from the corresponding author on reasonable request.

**Acknowledgments:** We thank Jeffrey Long for reviewing the manuscript.

**Conflicts of Interest:** Philibert is the Chief Executive Officer of Behavioral Diagnostics. The use of cg05575921 to assess smoking status is covered by existing and pending patents including US Patents 8,637,652 and 9,273,358. The University of Iowa has filed intellectual property claims related to the integrated genetic/epigenetic technology described in this communication (US Patent application 62,455,416: Compositions and Methods for Detecting Predisposition to Cardiovascular Disease) on behalf of Dogan and Philibert. Notice of intent for allowance for this application has been made by the European Patent Office. In addition, Dogan and Philibert, Cardio Diagnostics Inc., have filed intellectual property claims related to the integrated genetic/epigenetic technology described in this communication (US Patent application 63,074,878: Methods and Compositions for Predicting Coronary Heart Disease). Dogan is the Chief Executive Officer and stockholder of Cardio Diagnostics Inc.; Philibert is the Chief Medical Officer and stockholder of Cardio Diagnostics Inc. (www.cardiodiagnosticsinc.com, accessed on 6 April 2021). The other authors have no applicable conflicts.

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
