# Peer review of "The Reversion of DNA Methylation at Coronary Heart Disease Risk Loci in Response to Prevention Therapy"

_processes, doi:10.3390/pr9040699_

Round 1

Reviewer 1 Report

Dear Author,

Thank you for submitting manuscript titled “Reversion of DNA Methylation at Coronary Heart Disease Risk Loci Demonstrates the Potential of Epigenetics for Guiding Coronary Heart Disease Prevention

First of all, congratulation of this novel study and I respect of your great work. I have carefully read and evaluated this manuscript with great interests, and I have several minor comments for acceptance of Processes.

Thank you for your consideration. If you have any questions and problems, please feel free to ask me ASAP.

Reviewer comments

This manuscript is well written about the novel study to assess the epigenetics approach for analysis of risk factors for coronary heart disease. This technique seems to be quite unique and still hard to understand in detail. But, I believe that this novel technique will be of great importance for the future analysis of CAD. However, some additional information will be needed to elucidate the precise mechanism. Please revise this manuscript following these comments as below.

Minor comments:

  1. Table 1: Please show the p-value in comparison of both groups.
  2. Figure 1: Too small to understand at a glance. Please make more clear.
  3. In discussion, the limitation of this study will be place the end of discussion.
  4. References: The style of references should be checked and revised following the “Submission Guideline” of this journal. The name of Journal should be an abbreviation.
  5. Institutional Review Board Statement: ap-proved → approved

Author Response

Comment: “Table 1: Please show the p-value in comparison of both groups.”
Response:  Done per the Reviewer’s request.  Interestingly, there were many group comparison p-values in the trend range (0.1< p >0.05). But only one comparison was significant, and that was before correction for multiple comparisons.

Comment: “Figure 1: Too small to understand at a glance. Please make more clear.”
Response: We have revised Figure 1.

Comment: “In discussion, the limitation of this study will be place the end of discussion”
Response:  We have moved the limitations section to the end of the discussion per the Reviewer’s request.

Comment: “References: The style of references should be checked and revised following the “Submission Guideline” of this journal. The name of Journal should be an abbreviation.”
Response:  Fixed per the Reviewer’s request.  All journal titles are now according to the ISO4 abbreviation standard.  A duplicate reference was also removed.

Comment: “Institutional Review Board Statement: ap-proved → approved”
Response:  “We fixed the typo. Thank you for spotting it.

Sincerely,

Robert A. Philibert M.D., Ph.D.
Professor of Psychiatry and Biomedical Engineering
Member, Neuroscience and Genetics Programs

Reviewer 2 Report

Thank you for the opportunity to review your paper. The manuscript could be improved by reporting exact p-values instead of using "<" sign. Please, remove all parts where your saying that there is a trend. Maybe you could also re-evaluate the title because as such, it is a an overstatement. 

Author Response

Comment: “The manuscript could be improved by reporting exact p-values instead of using "<" sign”
Response:  Exact p-values are for comparison of categorical variables.  Methylation is a continuous variable.  Hence the use of the standard p-values is appropriate.

Comment: “Please, remove all parts where your saying that there is a trend.”
Response:  Done as requested by the Reviewer.

Comment: “Maybe you could also re-evaluate the title because as such, it is an overstatement.”
Response:  We have modified the title per the Reviewer’s Request.

Sincerely,

Robert A. Philibert M.D., Ph.D.
Professor of Psychiatry and Biomedical Engineering
Member, Neuroscience and Genetics Programs

Reviewer 3 Report

This team previously reported methylation of CHD associated loci. Here they furthered this finding by investigating influence of smoking cessation (i.e. primary prevention) on DNA methylation on these loci and reported alterations in certain loci following smoking cessation. As they discussed, the lower levels of methylation make it unlikely to influence expression of regional genes. Nonetheless, the finding, albeit premature as a pilot study, is potentially important and would stimulate further research.

I have a few comments to this work.

  1. The introduction reads excessively long. The authors are suggested to remove some irrelevant discussion on likely intervention on primary risk factors of CHD to make this paper concise and more focused.
  2. It remains unknown on the turnover of DNA methylation following an intervention. You indicated that subjects completed 4 visits over a 3-month period. It would be interesting to determine using blood samples collected during these visits to determine anges in CHD-associated methylation in the middle of the 3-month period. This additional data, together with the results from the end of the study period, might allow for assessment of the turnover rate of DNA-methylation
  3. Table: a key index missing in the Table smoking history (i.e. year of smoking), which should have been presented and taken into analysis. Any difference in the duration of smoking between the two groups?

Author Response

Comment: “The introduction reads excessively long. The authors are suggested to remove some irrelevant discussion on likely intervention on primary risk factors of CHD to make this paper concise and more focused.”
Response:  We have removed 5 lines of  material on primary prevention per the Reviewer’s request. But we note that setting context of the material is important here.

Comment: “It would be interesting to determine using blood samples collected during these visits to determine changes in CHD-associated methylation in the middle of the 3-month period.”
Response:  We agree that it would be interesting but it would also require additional laboratory work and a complete redo of the analytic approach. The real question that the Reviewer wishes to know is the same that we wish to know-what happens at 6 months and one year.  That study will follow on the heels of this pilot study.  But it will be an expensive study to perform because of the need to monitor treatment compliance.

Comment: “Table: a key index missing in the Table smoking history (i.e. year of smoking), which should have been presented and taken into analysis. Any difference in the duration of smoking between the two groups?”
Response:  The duration of smoking is for each group is expressed as “pack years consumption” and was found not to be different between the two groups (see Table 1).  The reasons for this are complex and are due in part to the large standard deviation in the non-quitting group. But the objective marker cg05575921 assesses current smoking intensity quite well with this paper shows the utility of using objective markers when evaluating changes in cardiac risk.  

Sincerely,

Robert A. Philibert M.D., Ph.D.
Professor of Psychiatry and Biomedical Engineering
Member, Neuroscience and Genetics Programs

Round 2

Reviewer 3 Report

The authors replied to my concerns that are acceptable, given that the nature of this study was a preliminary one.

I have no further comments.